# Stimulation of Osteoclast Formation by Oncostatin M and the Role of WNT16 as a Negative Feedback Regulator

**DOI:** 10.3390/ijms23063287

**Published:** 2022-03-18

**Authors:** Pedro P. C. de Souza, Petra Henning, Ulf H. Lerner

**Affiliations:** 1The Innovation in Biomaterials Laboratory, School of Dentistry, Federal University of Goiás, Goiânia 74690-900, Brazil; pedrosouza@ufg.br; 2Department of Internal Medicine and Clinical Nutrition, Institute of Medicine, Sahlgrenska Osteoporosis Centre and Centre for Bone and Arthritis Research at the Sahlgrenska Academy, University of Gothenburg, 413 45 Gothenburg, Sweden; petra.henning@gu.se

**Keywords:** OSM, WNT16, RANKL, osteoclast, bone resorption

## Abstract

Oncostatin M (OSM), which belongs to the IL-6 family of cytokines, is the most potent and effective stimulator of osteoclast formation in this family, as assessed by different in vitro assays. Osteoclastogenesis induced by the IL-6 type of cytokines is mediated by the induction and paracrine stimulation of the osteoclastogenic cytokine receptor activator of nuclear factor κ-B ligand (RANKL), expressed on osteoblast cell membranes and targeting the receptor activator of nuclear factor κ-B (RANK) on osteoclast progenitor cells. The potent effect of OSM on osteoclastogenesis is due to an unusually robust induction of RANKL in osteoblasts through the OSM receptor (OSMR), mediated by a JAK–STAT/MAPK signaling pathway and by unique recruitment of the adapter protein Shc1 to the OSMR. Gene deletion of *Osmr* in mice results in decreased numbers of osteoclasts and enhanced trabecular bone caused by increased trabecular thickness, indicating that OSM may play a role in physiological regulation of bone remodeling. However, increased amounts of OSM, either through administration of recombinant protein or of adenoviral vectors expressing *Osm*, results in enhanced bone mass due to increased bone formation without any clear sign of increased osteoclast numbers, a finding which can be reconciled by cell culture experiments demonstrating that OSM can induce osteoblast differentiation and stimulate mineralization of bone nodules in such cultures. Thus, in vitro studies and gene deletion experiments show that OSM is a stimulator of osteoclast formation, whereas administration of OSM to mice shows that OSM is not a strong stimulator of osteoclastogenesis in vivo when administered to adult animals. These observations could be explained by our recent finding showing that OSM is a potent stimulator of the osteoclastogenesis inhibitor WNT16, acting in a negative feedback loop to reduce OSM-induced osteoclast formation.

## 1. Introduction

The skeleton, together with muscles attached to its surfaces, plays a crucial role in movement, but is also involved in the homeostatic mechanisms tightly controlling calcium concentrations in serum. Bone tissue also harbors hematopoietic bone marrow and bone cells play a role in hematopoiesis in different bone marrow niches. In addition, mineralized bone serves as protection for certain internal organs such as brain, lungs and heart.

Skeleton contains cortical bone, which forms a shell in the periphery and trabecular bone residing on the inside as a honeycomb-like network. The tissue consists of an organic matrix, mainly of collagen type I fibers, but also of several non-collagen proteins, which together serve as a scaffold for the inorganic part consisting of hydroxyapatite crystals. 

The cells forming bone matrix and regulating hydroxyapatite precipitation are osteoblasts, which are mesenchymal-derived cells, as are the closely related chondrocytes and adipocytes [1]. Terminally differentiated, bone-forming osteoblasts are cuboid cells, which reside on bone surfaces in trabecular and cortical bone. Some of the osteoblasts are transformed to a thin layer of resting cells, so called bone-lining cells, and others are incorporated into bone and transformed to osteocytes with many cell extensions, which make contact with osteoblasts on surfaces and other osteocytes in bone. Osteocytes are by far the most abundant bone cell type [2] and are important to sensing mechanical load on the skeleton and thereby taking part in regulation of bone mass [3]. Osteocytes also play a role in the initiation of resorption of old bone in the remodeling process [4,5]. 

The cells resorbing bone in physiological and pathological processes are osteoclasts, which are multinucleated giant cells derived from mononucleated, myeloid, hematopoietic stem cells with a unique ability to resorb bone [6,7,8]. Similar to osteoblasts, mature osteoclasts are present on surfaces on bone, but can also be found in remodeling Haversian canals within cortical bone [9]. Osteoclasts attach to bone surfaces through a sealing zone, which surrounds the resorption area, the so called Howship’s resorption lacunae. In the lacunae, the hydroxyapatite crystals are dissolved by acid produced through a proton pump and a chloride channel expressed in the osteoclast cell membrane towards the resorption lacuna. The demineralized organic bone matrix is then degraded by a cocktail of proteolytic enzymes. See below for further details regarding how osteoclast formation is controlled.

The amount of bone tissue and the structure of bones are determined by two processes, bone modeling and bone remodeling [10]. Bone modeling is important for the shape of bone, whereas bone remodeling does not change the shape, but is important for renewing old, damaged bone with new healthy bone tissue. Bone modeling is particularly important during skeletal development to produce bones with a form and structure adopted to resist load (Figure 1A), but can also be observed in pathological conditions such as inflammation close to bone in rheumatoid arthritis and periodontitis (Figure 1B, right) and bone loss in the vicinity of tumor metastases. Bone remodeling sites (Figure 1B, left), known as bone multicellular units (BMU), are present simultaneously at very many different sites at surfaces of trabecular and cortical bones, as well as in Haversian canals in cortical bone. 

The systemic and local factors controlling bone modeling during skeletal development and growth are to a large extent unknown, although sex steroid hormones produced during puberty are known to be important [11]. Bone modeling in pathological conditions is likely to be induced locally by inflammatory processes and tumor cells and is mediated by several cytokines known to affect osteoblasts and osteoclasts. 

Physiological remodeling occurs asynchronously and therefore needs to be regulated locally, but may also depend on systemic factors [10]. In remodeling, much attention has been paid to the processes by which the subsequent new bone formation is initiated in the resorption lacuna, a process known as coupling between resorption and formation. A wide variety of potential coupling factors have been described, some of them are bone matrix derived and therefore dependent on the resorption activity, and others are resorption-independent osteoclast secreted factors, factors expressed on osteoclast cell membranes, or factors dependent on release of microvesicles from osteoclasts [10].

Participation in physiological bone remodeling by cytokines in the IL-6 family such as oncostatin M (OSM) is suggested by findings showing that global or bone cell-specific deletion of several of the ligands, their cognate receptors, or the gp130 receptor component crucial for intracellular signaling induced by these cytokines results in skeletal phenotypes [12]. The list of the bone-active IL-6-type of cytokines includes IL-6, IL-11, leukemia inhibitory factor (LIF), OSM, cardiotrophin-1 (CT-1), ciliary neurotrophic factor (CNTF), IL-27 and IL-35 [12,13]. LIF, OSM and CT-1 can be produced and released by osteoclasts and may play a role in the coupling process by which osteoclasts recruit osteoblasts during bone remodeling [10]. Overexpression of *IL-6*, *IL-11* and *Osm*, and of hyperactive *gp130*, also results in skeletal phenotypes, indicating that these cytokines may play a role in bone modeling in pathological conditions [13]. The effects of all the IL-6 type of cytokines on bone resorption and bone formation are described in an excellent recent review and are summarized in a separate section below [13].

The present overview will describe how bone mass is regulated by OSM in physiological and pathological conditions. Although bone mass is dependent on both bone resorption and bone formation, the overview will mainly describe effects of OSM on bone resorption, including our recent finding of the role of OSM-stimulated osteoblast-derived WNT16 acting in a negative feedback loop to restrict the stimulatory effect of OSM on osteoclast formation [14]. 

## 2. Osteoclast Formation

As described above, osteoclasts are derived from myeloid, hematopoietic cells and are most closely related to monocytes/macrophages and dendritic cells in the immune system. For proliferation and survival of progenitor cells, cytokine macrophage colony-stimulating factor (M-CSF), also known as colony-stimulating factor-1 (CSF-1), is crucial [15,16,17]. Surprisingly, IL-34 can also bind to the M-CSF receptor [18] and substitute for M-CSF during osteoclastogenesis [19]. The relative importance of M-CSF and IL-34 as activators of cFMS/CSF1R in physiological remodeling of bone or in pathological processes affecting bone modeling is currently not known.

Crucial for osteoclast differentiation is receptor activator of nuclear factor κ-B ligand (RANKL) expressed on the surface of osteoblasts/stromal cells and its cognate receptor activator of nuclear factor κ-B (RANK) expressed on osteoclast progenitor cells [20]. The interaction between RANKL and RANK is inhibited by the soluble, decoy receptor osteoprotegerin (OPG), which similarly to RANK is a member of the same TNF receptor superfamily, and interferes with the interaction by binding to RANKL. The importance of the RANKL–RANK–OPG system in bone was initially shown by the observation that deletion of the gene (*Tnfrsf11b*) encoding OPG resulted in early-onset low bone mass (osteoporosis) in mice caused by excessive amounts of osteoclasts [21]. Later, it was demonstrated that global deletion of the genes encoding either RANKL (*Tnfsf11*) or RANK (*Tnfrsf11a*) resulted in mice with high bone mass due to lack of mature osteoclasts [22,23]. 

Although OPG is ubiquitously expressed, osteoblast-specific deletion of OPG has shown that it is locally, rather than systemically, produced OPG that is important for regulation of bone mass [24,25]. Numerous studies showed that not only parathyroid hormone (PTH) and 1,25(OH)_2_-vitamin D3, but most stimulators of osteoclast formation, initiate the differentiation of monocytes/macrophages along the osteoclastic lineage by increasing RANKL and/or decreasing OPG [26]. In the list of molecules capable of regulating RANKL/OPG are cytokines in the IL-6 family including OSM [12], which will be discussed in detail below.

The findings showing the crucial role of the RANKL–RANK–OPG system for bone mass were preceded by observations made by immunologists showing that interactions between RANK, expressed on dendritic cells, and TRANCE (RANKL) on T cells are important for adaptive immunity [27,28]. Accordingly, RANK- or RANKL-deficient mice also exhibit abnormal development of thymus and lymphoid tissues. Following these discoveries, it has been shown that RANKL is also expressed in mammary epithelial cells, hypothalamus, synovial fibroblasts, T cells, B cells and NK cells [20]. 

Activation of the trimeric RANK complex on lineage-committed mononuclear progenitor cells causes activation of a cascade of downstream signaling events [29], including recruitment of TNF receptor associated factor 6 (TRAF6) and subsequent activation of mitogen-activated protein kinase (MAPK) and transcription factors such as nuclear factor κB (NF-κB) and c-Fos containing activator protein-1 (AP-1) (Figure 2). Osteoclast differentiation also requires activation of immunoglobulin-like receptors on osteoclast progenitor cells such as Triggering receptor expressed on myeloid cells 2 (TREM-2) and osteoclast associated receptor (OSCAR), which associate with the receptor components DNAX activation protein of 12kDa (DAP12) and FcRγ, causing increased intracellular concentration of calcium required for expression, translocation and sustained activation of the transcription factor NFATc1, which is the key regulator of osteoclastogenesis [30]. RANK-induced activation of osteoclast differentiation also includes Blimp1-mediated transcriptional repression of anti-osteoclastogenic transcription factors important for macrophage differentiation such as Interferon regulatory factor 8 (Irf8) and MAF BZIP transcription factor B (MafB) [31]. 

In pathological conditions, RANKL expressed by cells other than osteoblasts/osteocytes can be important for osteoclast activation and bone loss. Thus, RANKL expressed by synovial fibroblasts causes articular bone erosions in mice with experimentally induced arthritis [32], whereas RANKL expressed by bone marrow plasma cells is important for periarticular bone loss [33]. In periodontal disease, it has been found that RANKL expressed in periodontal ligament cells is involved in the alveolar bone loss adjacent to teeth [34].

## 3. Bone Cell Expression and Binding Specificity of OSM, OSMR and LIFR

In 1986, Zarling et al. demonstrated that the supernatant collected from U-937 cells, a histiocytic lymphoma cell line, treated with phorbol 12-myristate 13-acetate (PMA), could inhibit the growth of A375 melanoma cells. The factor responsible for this effect was purified and named oncostatin M due to its capacity to inhibit the growth of cancer cells [35]. Since then, dysregulation of OSM production has been demonstrated in different types of cancer, such as myeloma [36], colon cancer [37] and pancreatic cancer [38]. However, expression of OSM is also associated with non-oncologic inflammatory diseases such as inflammatory bowel disease [39], COVID-19 [40], rheumatoid arthritis [41] and periodontal disease [42]. Thus, inflammatory cells such as neutrophils [43], dendritic cells [44], T cells [45] and macrophages [46] are major sources of OSM, but other cells including osteoblasts and osteocytes also express OSM [47].

Among the members of the IL-6 family, LIF is the most structurally, genetically and functionally similar cytokine to OSM [48]. Structurally, both cytokines are formed by a secondary structure containing four helical bundles, which are linked together, a structural feature that is common also for other members of the IL-6 family [49]. Genetically, the close gene location in the chromosome suggests that the genes encoding OSM and LIF were originated by duplication [50]. Due to these similarities, the signaling pathways activated by LIF and OSM present with some redundancies.

OSM binds to a heterodimeric receptor consisting of either gp130:OSMR (type II complex) or to the type I complex consisting of gp130:LIFR, the latter with lower affinity, in both mouse and human cells, while LIF binds exclusively to the gp130:LIFR heterodimer [48,51,52,53,54,55]. Interestingly, while gp130 functions as a β-receptor for most of the cytokines in the Il-6 family, being recruited only after binding of the cytokine to its respective α-receptor (i.e., IL-6 binds first to IL-6R or soluble IL-6R, LIF to LIFR and IL-11 to IL-11R, and the complexes are then recruited to gp130), it functions as a low-affinity α-receptor for OSM, which bind to gp130 before being recruited to OSMR or LIFR [48,51,52,53,54]. Because gp130 is ubiquitously expressed, the expression of OSMR and LIFR will determine cell responsiveness to OSM. Several bone cell types express OSMR and LIFR. In mice, OSMR is expressed in bone marrow stromal cell osteoblasts, bone-lining cells and osteocytes, but not in osteoclasts [47] (Figure 3). Likewise, LIFR expression has also been demonstrated in a variety of bone cells such as bone marrow stromal cells [56], pre-osteoblasts [57], osteoblasts [56,58,59], bone-lining cells and osteocytes [47]. LIFR is also expressed on macrophages, but not on osteoclasts [47], a finding explained by our observation showing that the LIFR in macrophages is downregulated when osteoclast formation is stimulated with RANKL [60]. Interestingly, expression of all receptor components—gp130, OSMR and LIFR—is augmented in the synovial tissue in a model of mouse antigen-induced arthritis [61]. 

The common use of the type I complex by OSM and LIF may be explained by similarities of the two cytokines revealed by computational modeling of their interaction with LIFR [62] based on existing data from crystallography studies of LIF–LIFR interaction [63] and site-directed mutagenesis of OSM [64,65]. Although the percentage of matching amino acid residues between human OSM and human LIF is only between 22 and 29% [66], they have convergent 3D structures comprising 4α-helices, named A, B C and D, linked by polypeptide loops [51,67]. Similarities in amino acid residues within the so called binding site III, comprising the AB loop, BC loop and D helix, as revealed by superposition of their crystal structures, may account for the common use of LIFR by OSM and LIF. This was further evidenced by protein–protein docking and molecular dynamics simulations [62]. Importantly, based on these docking experiments, the stability of the OSM:LIFR complex appears to be less stable than the LIF:LIFR complex. Interestingly, although human OSM (GenBank: CAG30420.1) and mouse OSM (GenBank: AAH99866.1) share 43% identity and 53.4% similarity (EBLOSUM62 Matrix, Gap penalty: 12, Extend penalty: 2), specific amino acids in the AB loop determine species specificity of mouse and human OSM and they do not bind OSMR from other species. While human OSM acts only through the type I complex in mouse, the mouse OSM does not activate any of the human receptor complexes [65].

Details on the signaling mechanisms after OSM binding to these receptors will be discussed later in this review. 

## 4. Stimulation of Osteoclastogenesis by OSM through RANKL

Tamura et al. reported for the first time that OSM (0.01, 0.1 and 1 ng/mL) increases osteoclast formation in co-cultures of mouse calvarial osteoblasts and bone marrow cells [68]. LIF was considerably less potent and caused a weak response only at 100 ng/mL. In co-cultures of the murine stromal cell/osteoblast cell line UAMS-32 and bone marrow stromal cells, it was then observed that mouse OSM (25 ng/mL) stimulated the expression of *Rankl* mRNA and formation of bone-resorbing osteoclasts [69].

In co-cultures of primary mouse calvarial osteoblasts and mouse bone marrow cells, mouse OSM (1.25–10 ng/mL) increased the formation of osteoclasts capable of making resorption pits in dentine slices [70]. Additionally, mouse LIF, used at the same concentrations, increased the numbers of bone-resorbing osteoclasts in these cultures, but the response was considerably less. In these cultures, the stimulatory effect of OSM on osteoclastogenesis was decreased by the non-steroidal anti-inflammatory compound indomethacin, indicating that the response was dependent on prostaglandin biosynthesis, most likely by the osteoblasts, which are known to produce prostaglandins in response to many stimuli [71]. Additionally, steroidal anti-inflammatory compounds such as glucocorticoids are potent inhibitors of prostaglandin biosynthesis. However, dexamethasone did not inhibit the osteoclastogenesis response to OSM, but on the contrary potentiated the effect [70]. Although the mechanism by which dexamethasone potentiated the effect of OSM was not demonstrated, the same was true also for the response to LIF. 

In ex vivo cultures of neonatal mouse calvarial bone, mouse OSM, human OSM and mouse LIF stimulate bone resorption, but mouse OSM is more effective than human OSM and mouse LIF [58]. The effect of mouse OSM was associated with increased mRNA and protein expression of RANKL. 

Using cells from mice in which *Osmr* had been globally deleted, Walker et al. showed that the osteoclastogenic response to mouse OSM in co-cultures of primary calvarial osteoblasts and bone marrow macrophages is due to activation of OSMR in osteoblasts [47], a finding in agreement with the fact that osteoblasts, but not osteoclasts express *Osmr* mRNA [60]. Similar to the observations in calvarial bones, mouse OSM induced the mRNA expression of *Rankl* in primary mouse calvarial osteoblasts and in the mouse stromal cell line Kusa4b10. This response to mouse OSM was absent in osteoblasts from *Osmr*^−/−^ mice [47], or in some experiments weak since a considerably delayed but small increase in *Rankl* expression was reported in osteoblasts from *Osmr*^−/−^ mice [72], which is in agreement with the observation that the osteoclastogenic effect of OSM is dependent on OSMR in osteoblasts. In contrast, human OSM and mouse LIF increased the mRNA expression of *Rankl* in *Osmr*^−/−^ osteoblasts, indicating that both these cytokines were using LIFR. Interestingly, mouse OSM, similar to human OSM and mouse LIF, decreased the expression of *Sost* mRNA (encoding the WNT inhibitor sclerostin) in osteoblasts from *Osmr*^−/−^ mice, showing that mouse OSM can also utilize mouse LIFR for some of its effects in bone. Immune histochemical evidence shows that these cytokines also decrease protein expression of sclerostin in osteocytes, subsequently causing increased WNT canonical signaling and enhanced bone formation [47]. Thus, it seems mouse OSM utilize OSMR to stimulate bone resorption and LIFR to enhance bone formation. 

In conditions with low serum calcium, PTH is well known as a physiological stimulator of bone resorption to increase serum calcium concentrations. Increased resorption is due to PTH-induced increased expression of RANKL in osteoblasts through PTH1 receptors. Pathologically, high PTH concentrations during hyperparathyroidism cause extensive bone resorption. However, intermittently administered PTH to increase bone formation is used as a bone anabolic treatment in patients with osteoporosis [73]. Based upon the observations that OSM can stimulate bone formation and that PTH increases the mRNA expression of *Osmr* and *IL-6st* (encoding gp130) in primary mouse calvarial osteoblasts and in Kusa4b10 cells [74], it has been assessed if OSM mediates the bone anabolic effect of PTH through OSMR. It was then found that absence of OSMR changed the anabolic response to intermittent PTH to a catabolic response, causing decreased trabecular bone mass in *Osmr*^−*/*−^mice, instead of the increased bone mass observed in wild-type mice [74]. The PTH-induced enhanced bone formation was similar in *Osmr*^−/−^ and wild-type mice, but the decreased bone mass was caused by increased numbers of osteoclasts in *Osmr*^−/−^ mice [74]. This was because PTH causes a more sustained expression of *Rankl* mRNA in *Osmr*^−/−^ osteoblasts compared to wild-type cells. The physiological or pathophysiological implication of this interesting observation is, however, currently not known.

Although we showed in a recent study that both OSM and LIF stimulate bone resorption in neonatal mouse calvariae, as assessed by release of prelabelled ^45^Ca, OSM is considerably more potent and effective, stimulating ^45^Ca release at ≥0.3 ng/mL and causing a maximal 2-fold increase at 30 ng/mL, whereas LIF caused a 1.3-fold increase at ≥3 ng/mL [60]. In agreement with these observations, OSM caused a clearly more robust increase in *Rankl* mRNA and protein expression in the calvariae. Similar results were found in mouse calvarial osteoblasts, where OSM caused a robust increase in *Rankl* mRNA expression and LIF caused only a weak response. Experiments in which OSM and LIF receptors were downregulated using a small interfering RNA technique demonstrated that the effect of OSM on *Rankl* mRNA expression was independent of LIFR, but dependent on OSMR, similar to observations using osteoblasts from *Osmr*^−*/*−^ mice [47,60]. In contrast, OSM, but not LIF, increased osteoclast formation and *Rankl* mRNA expression in crude bone marrow cell cultures, in agreement with that OSM, but not LIF, increased *Rankl* mRNA expression in the mouse bone marrow stromal cell line ST-2. 

In contrast to the studies showing that OSM stimulates osteoclast formation and bone resorption in cell and organ cultures, it has been reported that OSM, for reasons not understood, inhibits basal release of ^45^Ca from ex vivo cultures of fetal mouse long bones, without affecting PTH-stimulated release of ^45^Ca [75].

## 5. OSMR Signaling Inducing Osteoclast Formation

Among the factors controlling the fine tuning of bone resorption and formation, cytokines using gp130 as signal transducer deserve special attention due to their capacity to influence both osteoblasts and osteoclasts [7]. The essential role of gp130 for bone mass has been assessed in mutant mice in which the C-terminal moiety of gp130 was modified to selectively interfere in the STAT3 and the SHP-2/MAPK pathway [76]. In mice carrying a deletion in the binding and activation sites of STAT1/3 (*gp130*^Δ*STATΔSTAT*^), premature closing of the growth plate was observed, leading to reduced bone size, indicating that the STAT pathway is involved in chondrocyte proliferation and or differentiation. However, no changes in osteoblastic or osteoclastic activity, or trabecular bone volume, were found in these mice. On the other hand, mice carrying a substitution of Tyr^757^ with Phe^757^ (*gp130^F^*^757/*F*757^), which results in defective SHP-2/MAPK signaling, had increased bone turnover leading to an osteopenic phenotype. This was due to increased osteoclastogenesis in these mice, suggesting that the SHP-2/MAPK pathway activated by gp130 is an osteoclast inhibitory pathway. In another study, it was demonstrated that mice carrying the gp130*^F^*^759/*F*759^ mutation also showed increased bone volume, in this case partially due to hyperactivation of the STAT3 pathway [77]. These studies highlight the importance of gp130 for bone metabolism. Although the common use of gp130 by OSM and other members of the IL-6 family trigger some shared phenotypes, differences are also observed and may be explained by differences in signaling downstream to the different receptor components that associate with gp130. 

In the murine system, it was believed that mouse OSM was capable of binding with high affinity only to the gp130:OSMR complex [78], but the finding that OSM strongly suppresses sclerostin in osteoblasts from *Osmr*^−*/*−^ mice [47] suggests that some important effects of OSM may be due to activation of the type I receptor complex in this system. This seems not to be the case for the OSM-induced expression of RANKL as demonstrated by silencing experiments using calvarial osteoblasts [60]. In these cells, OSM strongly induces *Rankl* mRNA expression, an effect that is diminished by silencing of *IL-6st* and *Osmr* mRNAs using siRNAs, but not by silencing of *Lifr* [60]. 

Upon binding of OSM, the dimerization of the receptor subunits activates multiple signaling pathways, including the Janus kinase and signal transducer and activator of transcription (JAK–STAT) pathway, the mitogen-activated protein kinases ERK1/ERK2, the stress-activated protein kinases p38 and c-Jun N-terminal kinase (JNK), the phosphatidylinositol-3-kinase (PI3K)/Akt pathway and protein kinase C delta (PKCδ) [60,79]. This is due to the usage of multiple docking sites found in the cytoplasmic tail of gp130, LIFR or OSMR [80]. Thus, depending on which receptor complex OSM binds, and depending on the cell type, different downstream molecules may be recruited to propagate the intracellular signaling. In mouse calvarial osteoblasts, OSM activates ERK1/ERK2, JNK and STAT3, but not p38 [60]. Of all the pathways activated by OSM, the JAK–STAT pathway is considered the main mediator, as well as for other members in the IL-6 family [81]. 

The JAK-binding sites in gp130 are highly conserved regions that are present in several cytokine receptors, namely box 1 and box 2 [82]. These box regions are within an approximately 60 amino acid segment located proximal to the transmembrane domain. Box 1 comprises a cluster of hydrophobic amino acids followed by a proline-rich motif of eight amino acids and box 2 begins with a cluster of hydrophobic amino acids harboring some negatively charged amino acids and ends with positively charged amino acids [82]. JAK 1, JAK 2 and Tyk2 associate with gp130 and are phosphorylated in response to several members of the IL-6 family, including IL-6, LIF, OSM, IL-11 and CNTF [81,83,84,85]. OSMR also contains the box 1 and box 2 regions that associate with JAK 1 and JAK 2 [80,86]. However, studies using *Jak*-deficient cell lines demonstrate that OSM signaling is less dependent on JAK 1 than IL-6 signaling [86,87,88]. On the other hand, LIFR preferentially uses JAK1 [89,90] despite the capacity to also recruit JAK 2 and Tyk2 [85].

After association with JAKs, STATs are recruited to the receptor subunits, dimerize, and translocate to the nucleus to regulate the expression of target genes. Both STAT1 and STAT3 are phosphorylated by gp130, but they have different affinities for the conserved tyrosine residues in this receptor. STAT1 is recruited exclusively to Tyr^905^ and Tyr^915^ [91], while STAT3 associates with Tyr^767^ and Tyr^814^ besides its association with Tyr^905^ and Tyr^915^ [92]. Activation of gp130 also recruits the Src homology domain protein tyrosine phosphatase (SHP2) via the Src homology 2 domain [93,94]. This interaction is important for MAPK activation by IL-6 [95]. 

Upon heterodimerization with gp130 and association with JAKs, LIFR and OSMR also have the capacity to recruit STATs. Thus, OSM binding to the receptor complexes also induces signal propagation through the β-receptors. After its association with JAK 1, LIFR stimulates three signaling pathways—JAK–STAT [85], MAPK [96] and PI3-kinase [97]—depending on the cell type. The OSMR subunit activates STAT1 independently of binding to tyrosine motifs, most likely due to direct binding of STAT1 to JAK1 [86]. OSMR also recruits STAT3 through tyrosine motifs that are highly conserved among species, Tyr^917^ and Tyr^945^ [86,98], and STAT5 by direct recruitment to Tyr^837^ and Tyr^839^ in the human receptor or due to direct association with JAK2 in the mouse receptor [86]. Walker et al. showed that OSM increased the phosphorylation of STAT1, 3 and 5 in calvarial osteoblasts through OSMR [47]. Interestingly, treatment of calvarial osteoblasts with OSM stimulates a stronger phosphorylation of STAT3, ERK1/2 and JNK1/2 in comparison to LIF [60], in agreement with the fact that OSM is a more potent stimulator than LIF of osteoclast formation [60]. This may be explained by the presence of different adapter proteins that associate with OSMR, but not with LIFR. Activation of the STAT3 pathway in osteoblasts stimulates the expression of *Rankl* mRNA and osteoclast formation when osteoblastic cells are co-cultured with bone marrow cells [69]. OSM treatment also increases the expression of *Rankl* in bone marrow stromal ST2 cells, an effect mediated via increased binding of STAT3 and RNA polymerase II to distal enhancer regions upstream the *Rankl* gene transcription start site [99].

An important non-redundant pathway between OSM and the other IL-6 cytokines was identified in year 2000 by two independent groups [98,100]. It involves the recruitment of Shc1 specifically to the conserved Tyr^861^ of the OSMR subunit [98]. Shc1 belongs to a family of proteins comprising products from four different genes, named Shc1, Shc2, Shc3 and Shc4, or alternatively ShcA, ShcB, ShcC and ShcD [101,102]. Shc1 is ubiquitously expressed and exists in three isoforms with molecular masses of 46, 52 and 66 kDa [103,104]. In mouse osteoblasts, the three isoforms can be detected, but only *Shc1* mRNA expression is regulated by OSM [60]. In contrast, LIF had no effect on any of the isoforms. Interestingly, Shc1 is phosphorylated by OSM and silencing of *Shc1* using siRNA reduced STAT3 phosphorylation and *Rankl* mRNA expression induced by OSM, suggesting that the unique stimulatory effects of OSM are due to recruitment of Shc1, which facilitates STAT3 phosphorylation, leading to increased expression of *Rankl*. Furthermore, osteoblasts that were knocked down for *Shc1* had impaired capacity to induce osteoclastogenesis when they were cocultured with bone marrow macrophages [60]. In this study, the selected siRNA targeted all three isoforms of *Shc1*, making it impossible to analyze the role of each specific isoform in OSM-induced RANKL expression. Thus, studies inducing overexpression or downregulation of the specific isoforms could further elucidate the role of Shc1 in OSM-induced effects in osteoblasts.

In summary, both the type I and II receptor complexes can be activated by OSM in the murine system. Signaling downstream by both complexes involves activation of multiple pathways, including JAK–STAT, ERK and JNK. The expression of *Rankl* induced by OSM is strongly triggered by activation of the gp130:OSMR heterodimer due to the exclusive recruitment of Shc1 to the OSMR subunit, which results in enhanced phosphorylation of STAT3 to robustly induce *Rankl* mRNA expression (Figure 4).

## 6. Effects of OSM on Bone In Vivo

Although most observations made in cell and organ cultures demonstrate that OSM is a robust stimulator of osteoclast formation and bone resorption, the in vivo studies are more inconsistent.

Global deletion of *Osmr* in mice results in increased trabecular bone mass, which was associated with both decreased numbers of osteoclasts and decreased bone formation [47]. These observations indicate that OSM is a physiological stimulator of both bone resorption and bone formation. Further evidence for a physiological role in osteoclast formation was the observation that there were more remnants of cartilage in trabecular bone in primary spongiosa in *Osmr*^−/−^ mice.

Excessive amounts of OSM in vivo caused by injections of adenoviral vector-expressing *Osm* in knee joints result in increased numbers of osteoclasts and enhanced juxta-articular bone loss caused by increased RANKL, which was considerably potentiated by co-administration of adenoviral vector expressing either *IL-1* or *TNF-α* [105,106]. Since the injections also cause synovial hyperplasia and inflammation, it is not known if the effects on osteoclasts are due to increased inflammation or a more direct effect of OSM on bone cells. 

Initial observations indicating that increased amounts of OSM in vivo is rather a stimulator of bone formation than of bone resorption were the studies demonstrating that overexpression of bovine *Osm* resulted in abnormal formation of immature bone in marrow cavities and excessive periosteal new bone formation [107]. Intra-tibial and intra-articular injections of adenoviral vectors expressing *Osm* also result in increased bone formation in mice [46,108]. Injections of recombinant mouse OSM in the periosteum covering mouse calvarial bones cause increased calvarial thickness due to enhanced numbers and activity of osteoblasts [47,109], likely due to a suppression of sclerostin expression in osteocytes [47]. Additionally, in vitro, OSM stimulates osteoblast differentiation of stromal cells at the expense of adipogenic differentiation [13].

It is apparent that although OSM seems to be a physiological stimulator of bone resorption, excess OSM is a more potent stimulator of bone formation than of bone resorption in vivo. It might be that OSM induces an inhibitor of either RANKL production or downstream events involved in RANKL-induced osteoclastogenesis, which is more prominent in vivo than in vitro.

## 7. WNT16 as an Inhibitor of OSM-Induced Osteoclast Formation 

We recently discovered a novel mechanism whereby OSM-induced osteoclast differentiation can be balanced. We identified OSM and IL-6 as very strong stimulators of *Wnt16* expression in primary mouse calvarial osteoblasts [14]. Human large-scale genome-wide association studies have identified the *WNT16* locus as the strongest determinant of cortical bone mass and susceptibility to forearm fractures [110,111]. We and others showed that WNT16 is an important regulator of osteoclast numbers in vivo through mechanistic translational studies in mice [112,113,114]. Mice with the *Wnt16* gene deleted have increased numbers of osteoclasts [112,113], while mice with osteoblastic overexpression of *Wnt16* have decreased osteoclast numbers [114]. Moreover, in vitro studies using cultures of human monocytes, mouse bone marrow macrophages and mouse spleen cells showed a direct inhibitory effect of WNT16 on RANKL-induced osteoclast differentiation [14,112]. In primary calvarial bone cell cultures containing osteoblasts and osteoclast progenitors, OSM, in addition to inducing *Rankl* mRNA expression and osteoclast differentiation, increased the expression of *Wnt16* mRNA [14]. We further demonstrated that OSM-induced *Wnt16* mRNA expression in calvarial bone cells was mediated by the OSMR/gp130/Shc1/STAT3 signaling pathway. In cultures of calvarial bone cells from *Wnt16*^−*/*−^ mice, OSM-induced osteoclastogenesis was enhanced, showing that WNT16 acted as a negative feedback regulator of osteoclast differentiation induced by OSM [14]. Interestingly, this negative feedback mechanism could not be seen in whole-bone-marrow cell cultures containing stromal cells and osteoclast progenitors. OSM also induced the expression of *Rankl* and osteoclast differentiation in whole-bone-marrow cell cultures, but the expression of *Wnt16* was not increased and OSM-induced osteoclastogenesis was similar in cultures from wild-type and *Wnt16*^−*/*−^ mice. An explanation for the lack of *Wnt16* induction by OSM in whole-bone-marrow cell cultures was offered by single-cell RNAseq analysis, showing that *Osmr* and *Wnt16* mRNA are expressed in different cell types in whole-bone-marrow cell cultures. The expression of *Wnt16* mRNA was considerably lower in whole-bone-marrow cell cultures than in calvarial bone cells and this, together with the lack of OSM-dependent regulation of *Wnt16*, might explain why no negative feedback on OSM-induced osteoclastogenesis could be observed in whole-bone-marrow cell cultures. WNT16 is primarily important for cortical bone in vivo and it will be interesting to investigate if the effect of OSM on bone in vivo is influenced by a WNT16-mediated negative feedback mechanism on osteoclastogenesis.

The presence of an OSM-induced osteoclast inhibitory factor was suggested by de Hooge et al. already in 2002 [108], and we now suggest WNT16 to be such a factor. We hypothesize that WNT16 is involved in shifting the balance between OSM-induced bone formation and bone resorption by inhibiting osteoclast differentiation, resulting in increased bone mass in vivo (Figure 5). Deletion of *Wnt16* in mice has also been shown to decrease bone formation and this stimulatory effect on bone formation by WNT16 might further add to the bone anabolic effect of OSM [113,115].

## 8. Effects of Other Members of the IL-6 Family on Bone Resorption

The initial evidence suggesting that cytokines in the IL-6 family can increase osteoclast formation were findings showing that IL-6 stimulates bone resorption in ex vivo cultures of fetal mouse metacarpals [116] and fetal mouse calvaria [117]. In a subsequent study, it was shown that IL-6 stimulates formation of bone-resorbing osteoclasts in co-cultures of mouse calvarial osteoblasts and bone marrow cells, but concomitant addition of soluble IL-6 receptor was required for this effect [68]. Although we showed that IL-6 stimulates bone resorption in neonatal mouse calvarial bone cultures, in agreement with observations in fetal mouse organ-cultured bone explants, addition of soluble IL-6 receptor is needed in this system [58]. 

The IL-6 receptor signaling complex is initially formed by binding IL-6 to cell membrane-bound IL-6R followed by recruitment of gp130 [118]. This trimer then forms a hexameric complex with another IL-6:IL-6R:gp130 trimer to induce downstream *cis*-signaling [119]. Interestingly, later studies have indicated that the assembly of this IL-6 signaling complex differs somewhat in mice and human cells [120]. Rather few cell types express a cell membrane-bound IL-6R and, in most cases, cells are stimulated by so called *trans*-signaling in which the hexameric complex is made up by IL-6 initially binding to a soluble IL-6R.

Antibodies neutralizing gp130 inhibit the bone resorptive response to IL-6/soluble IL-6R in neonatal mouse calvarial bones, but do not affect the bone resorptive response to OSM, which might be related to the fact that OSM binds to a monomeric gp130, whereas IL-6/soluble IL-6R binds to a homodimer of gp130 [58]. 

The osteoclastogenic stimulatory effect of IL-6/soluble IL-6R is dependent on enhanced formation of RANKL in both neonatal mouse calvarial bones [58] and in co-cultures of mouse bone marrow cells and the osteoblastic cell line UAMS-32, an effect dependent on JAK–STAT signaling [69].

Although in vitro experiments have demonstrated that IL-6 can stimulate osteoclast formation and bone resorption, in vivo experiments have generated conflicting results since overexpression of *Il6* has been found to cause either decreased [121] or increased [122] numbers of osteoclasts, whereas global deletion of *Il6* did not affect osteoclast numbers [123]. These observations question if IL-6 plays any role in physiological remodeling of the skeleton. It seems, however, that IL-6 plays an important role in pathological processes since treatment of mice with antibodies neutralizing IL-6 inhibits bone loss caused by ovariectomy [124]. Similarly, germline deletion of *Il6* makes mice resistant to ovariectomy-induced bone loss [123]. Interestingly, antibodies neutralizing specifically IL-6 *trans*-signaling inhibit trabecular, but not cortical bone loss in ovariectomized mice [125].

Similar to the IL-6 receptor, IL-11 receptor signaling is induced by a hexameric complex consisting of two IL-11:IL-11R:gp130 trimers. Additionally, similar to IL-6, IL-11 stimulates osteoclast formation in co-cultures of mouse bone marrow cells and mouse calvarial osteoblasts [68,126], as well as bone resorption in organ cultures of mouse calvarial bones [126,127]. The bone resorptive effect is dependent on RANKL [127] and associated with increased mRNA expression of *Rankl* [128].

Interestingly, PTH and 1,25(OH)_2_-vitamin D3 stimulate the mRNA expression of *Il11* in bone marrow stromal cells [74,126] and antibodies neutralizing IL-11 decreased osteoclast formation induced by these hormones in co-cultures of bone marrow cells and calvarial osteoblasts [126]. However, these antibodies did not inhibit bone resorption in the mouse calvarial bones stimulated by the same hormones [127]. 

In mice with global deletion of *Il11ra*, trabecular bone mass is increased in both males and females, which is partially explained by decreased numbers of osteoclasts in females, but not in males [129]. However, there were more remnants of cartilage in secondary spongiosa in both males and females, indicating decreased resorption also in males. This seemed due to a cell lineage-autonomous effect since numbers of osteoclasts were reduced in RANKL-stimulated bone marrow cell cultures. In mice with transgenic overexpression of *Il11*, both cortical and trabecular bone mass are increased, an effect not associated with any effect of osteoclast numbers but with increased numbers and activity of osteoblasts [130].

Deletion of *Il11ra* in mice does not affect bone loss caused by ovariectomy [129], in contrast to deletion of *Il6* [123]. However, treatment of ovariectomized mice with anti-IL-11 has been found to reverse the decrease in bone mass and increased numbers of osteoclasts [131]. 

Similar to IL-6, the observations in cell and organ cultures strongly suggest that IL-11 is a stimulator of osteoclast formation, but the in vivo findings are less conclusive.

Downstream signaling induced by CT-1 is dependent on formation of a complex between LIFR, a monomer of gp130 and an elusive specific receptor for CT-1. CT-1, similar to IL-6, OSM and LIF, stimulates osteoclast formation in co-cultures of mouse bone marrow cells and mouse calvarial osteoblasts [70]. In contrast, the numbers of trabecular osteoclasts are increased in mice with global deletion of *Ct1* and several of these osteoclasts were oversized, but with no change in numbers of nuclei per cell [132]. This may be due to an intrinsic effect since osteoclasts expressed CT-1 and the numbers of osteoclasts were enhanced in RANKL-stimulated bone marrow macrophage cultures from *Ct1*-deficient mice. CT-1 does not stimulate *Rankl* mRNA expression in the bone marrow stromal cell line Kusa4b10 and in primary mouse calvarial osteoblasts [14,132].

It seems most observations indicate that CT-1 is not a stimulator of osteoclast formation in vitro, in contrast to IL-6, IL-11, OSM and LIF, but may play a role in vivo to regulate osteoclast spreading.

CNTF, which acts through a receptor complex consisting of CNTFR, LIFR and gp130, is not much studied in relation to osteoclastogenesis, but it has been observed that CNTF does not affect *Rankl* mRNA expression in mouse calvarial osteoblasts [14] and that the number of osteoclasts is not changed in *Cntf*^−*/*−^ mice [133]. 

IL-27 is a heterodimeric cytokine consisting of IL-27p28 (IL-27α) and Epstein–Barr virus-induced gene 3 (EBI3 or IL-27β). Since these subunits are related to subunits making up IL-12, IL-27 is also a member of the IL-12 family. IL-27 binds to a complex of IL-27R and a homodimer of gp130. The observations that IL-27 does not affect mRNA expression of *Rankl* or RANKL protein expression in mouse calvarial osteoblasts [14,134] indicate that IL-27 is not a stimulator of osteoclast formation. In contrast, it has been observed that IL-27 inhibits osteoclastogenesis in M-CSF/RANKL-stimulated human and mouse bone marrow macrophages and in human CD14^+^ monocytes [135,136,137,138]. The response was less robust in the mouse cells, which is explained by the lower expression of *IL-27* in the mouse macrophages.

In line with the in vitro findings, IL-27 reduced the numbers of osteoclasts in joints in mice with collagen-induced arthritis [138] and deletion of *Il27r* results in increased numbers of juxta-articular osteoclasts in mice given articular injections of LPS [136] similar to observations by Li et al. showing less bone loss by IL-27 treatment in LPS-stimulated mice [139]. These effects may be due to a combined indirect effect of decreased inflammation and a direct effect of IL-27 on osteoclasts. Further evidence for an inhibitory effect of IL-27 on osteoclast formation is the finding that treatment with IL-27 decreases osteoclast numbers and bone loss in ovariectomized mice [134].

IL-35 is a heterodimeric cytokine consisting of IL-12p35, which is also a subunit in the heterodimeric IL-12 cytokine, and the EBI3 subunits (IL-27β chain), a subunit also present in the heterodimeric IL-27 cytokine. Similar to IL-27, IL-35 seems to be an inhibitor of osteoclast differentiation since IL-35 inhibits formation of osteoclasts in M-CSF/RANKL-stimulated cultures of human CD14^+^ monocytes, due to a decrease in RANK receptors [140]. IL-35 also inhibits formation of bone-resorbing osteoclasts in mouse bone marrow macrophage cultures stimulated with M-CSF and TNF-α as well as bone loss in vivo induced by supracalvarial injections of TNF-α in mice [141].

Similar to OSM and LIF, IL-6, IL-11 and CT-1 stimulate bone formation in vitro and in vivo [12].

## 9. Concluding Remarks

OSM has been suggested to regulate bone mass in skeletal diseases such as rheumatoid arthritis [41,142], periodontitis [42] and neurogenic heterotopic ossification [143]. OSM may also play a role in skeletal metastasis of breast cancer. Decreased expression of *Osm* in breast cancer cells decreases bone metastasis and loss of trabecular and cortical bone mass [144]. No formal assessment of osteoclast numbers in vivo was made, but the authors showed that OSM enhanced osteoclast formation in RANKL-stimulated cocultures of breast cancer cells and the pro-osteoclastic cell line RAW264.7, an effect that was mediated by OSMR on breast cancer cells and decreased expression of OPG. In line with these studies, it has recently been reported that OSM may play a stimulatory role in bone metastases of ER^+^ (estrogen receptor positive) breast cancer cells in which the OSMR is more highly expressed compared to the less aggressive ER^−^ cells [145]. In inflammatory diseases, OSM is proposed to function as a stimulator of osteoclast formation and bone loss, whereas OSM is suggested to be a stimulator of bone formation in heterotopic ossification. Experiments performed in cell and bone organ cultures have demonstrated OSM to be a robust stimulator of osteoclast formation, whereas studies in mice showed that OSM is mainly a stimulator of bone mass through activation of osteoblasts; the latter findings can be reconciled with observations in osteoblast cell cultures. The discrepancies between in vitro and in vivo findings of OSM being a stimulator of osteoclast formation can be explained by our recent finding that OSM is a strong stimulator of the osteoclast inhibitor WNT16 [14].

OSM is expressed by a variety of cells including macrophages, dendritic cells in the immune system, neutrophils, T cells, muscle cells, as well as in osteoblasts and osteocytes, and found to be involved in several physiological and pathological processes [12]. Although global deletion of *Osmr* has shown that OSM may play a stimulatory role in osteoclast formation in physiological remodeling, the role of OSMR in osteoblasts and osteocytes needs to be assessed in mice with osteoblast/osteocyte-specific deletion of *Osmr*, to gain more detailed knowledge of the role of OSM in physiological and pathological osteoclast formation. Such experiments would also give us more insight into the stimulatory role of OSM on bone formation in vivo. The puzzling finding of OSM being a much stronger stimulator of bone formation than bone resorption in vivo and the potential role of WNT16 in this process need to be studied in vivo in mice with deletion of *Wnt16* in osteoblasts.

## Figures and Tables

**Figure 1 ijms-23-03287-f001:**
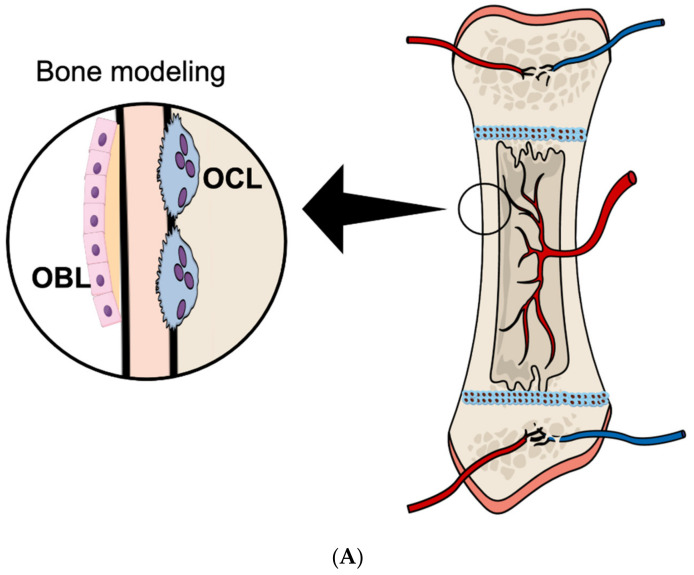
Bone modeling and remodeling: (**A**) During longitudinal bone growth, the diameter and thickness of the cortical bone are modeled to adjust to mechanical needs. To this end, osteoclasts (OCL) resorb bone in the endocortical region, while osteoblasts (OBL) deposit bone in the periosteal region. (**B**) In the alveolar bone surrounding the roots of the teeth, bone remodeling processes constantly occur to adapt to the mechanical needs of mastication (**left**). This process involves mechanical sensing by the osteocytes (OCY). Under conditions of inflammation, as in periodontitis, bone resorption is uncoupled from bone formation, resulting in bone modeling with loss of alveolar bone surrounding the root of the teeth (**right**).

**Figure 2 ijms-23-03287-f002:**
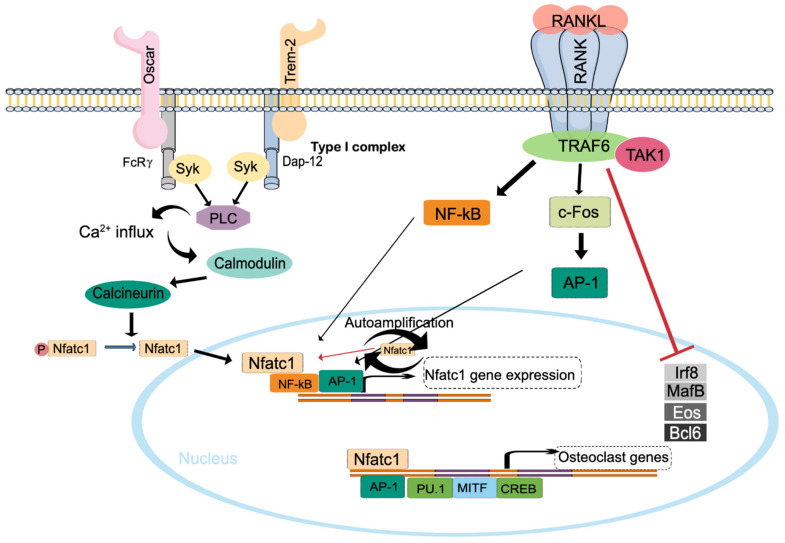
Osteoclastogenesis depends on the binding of RANKL to the trimeric receptor RANK and on co-stimulatory signals triggered by FcRy and Dap12 that dimerize with OSCAR and Trem-2, respectively. While RANK activation induces activation of important stimulatory transcription factors including CREB, AP-1, PU.1, Nfatc1, and MITF that regulate the expression of osteoclast genes, it downregulates other transcription factors such as Irf8, MafB, Eos and Bcl6. Meanwhile, signal through FcRy and Dap12 subunits activate PLC, which enhances intracellular calcium and thereby stimulates calmodulin and calcineurin. Calcineurin dephosphorylates Nfatc1, regarded as the master transcription factor for osteoclastogenesis, which translocates to the nucleus to autoamplify Nfatc1.

**Figure 3 ijms-23-03287-f003:**
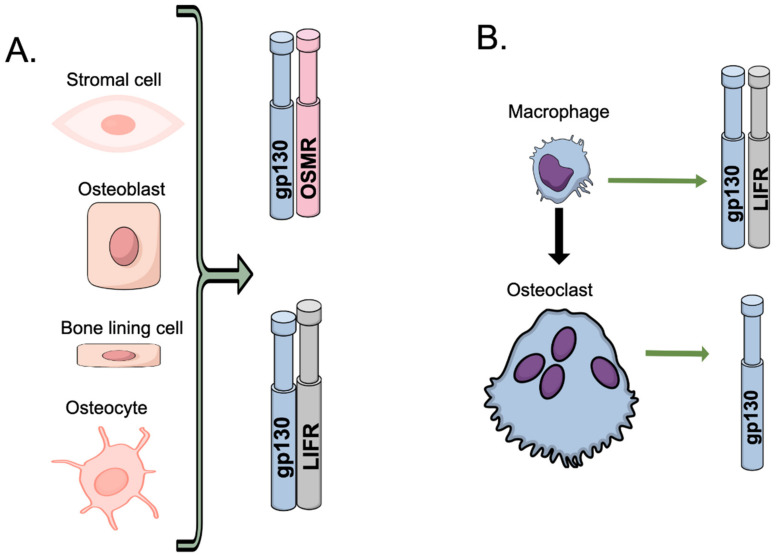
Expression of receptors recognizing OSM by bone cells. (**A**) Bone cells from mesenchymal origin, including bone marrow stromal cells, osteoblasts, bone-lining cells and osteocytes, express gp130, OSMR and LIFR. (**B**) Macrophages express gp130 and LIFR, but osteoclasts express only gp130, possibly because of downregulation of LIFR after treatment of macrophages with RANKL [60].

**Figure 4 ijms-23-03287-f004:**
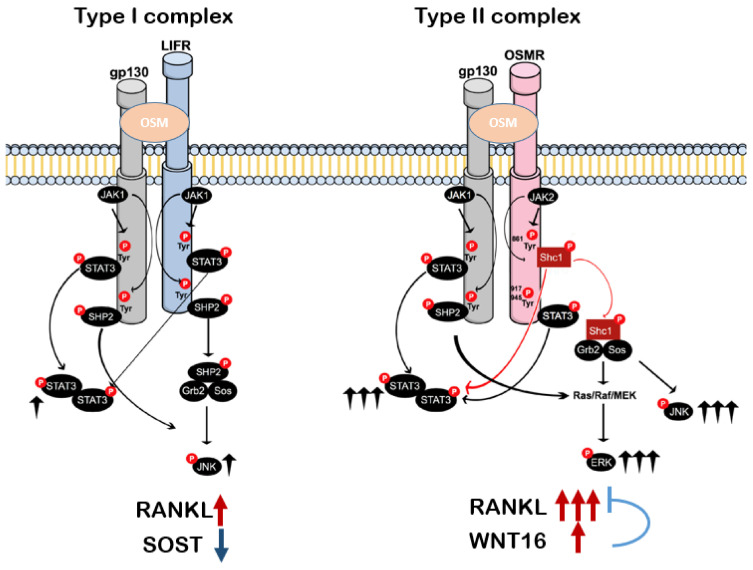
Schematic representation of OSM-induced signaling pathways in osteoblasts. OSM binding to the type I receptor complex activates STAT3 and JNK and regulates RANKL and sclerostin (SOST) expression. In addition to the gp130-mediated pathways common for LIFR and OSMR, the OSM receptor binding to the type II complex phosphorylates OSMR on Tyr^861^, resulting in docking and phosphorylation of the adapter molecule Shc1. The activated pShc1 is recruited to the Grb2:Sos complex, which in turn induces a Ras/Raf/MAPK cascade that ultimately activates ERK. The Shc1-mediated signaling pathway (**right**) is suggested to explain the stronger effects of OSM on expression of osteoclastogenic factors, osteoclast formation and bone resorption in comparison to activation of the LIFR:gp130 complex by LIF (**left**). This pathway also regulates Wnt16 expression that suppresses RANKL-induced osteoclastogenesis. Figure slightly modified from Persson et al. [60]. Copyright © 2019 Persson, Souza, Floriano-Marcelino, Conaway, Henning and Lerner. Licensed under a CC BY 4.0 license.

**Figure 5 ijms-23-03287-f005:**
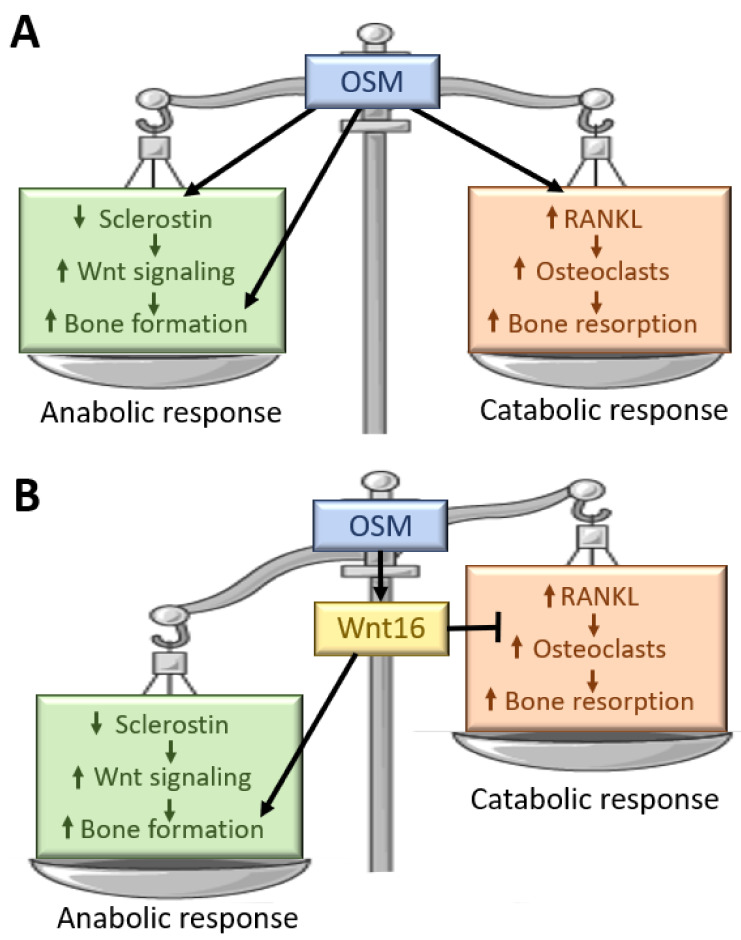
(**A**) OSM is believed to stimulate bone formation by decreasing sclerostin expression in osteocytes and thereby increasing WNT signaling and bone formation. Additionally, direct stimulatory effects of OSM on osteoblasts have been suggested by in vitro studies. However, OSM can also increase bone resorption by stimulating RANKL production and osteoclast formation. (**B**) OSM is a strong stimulator of WNT16 that has the capacity to both increase bone formation and inhibit osteoclast differentiation. Increased expression of WNT16 could shift the balance between bone anabolic and catabolic effects of OSM and be one factor explaining that most in vivo studies of OSM report increased bone mass.

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
