# Peer review of "Stimulation of Osteoclast Formation by Oncostatin M and the Role of WNT16 as a Negative Feedback Regulator"

_ijms, 2022, doi:10.3390/ijms23063287_

Round 1
Reviewer 1 Report
De Souza et al provide a comprehensive review of how oncostatin M stimulates osteoclast formation. While there is thorough coverage of the literature related to OSM, substantial changes to the text are needed. Predominantly, Sections 1 and 2 need to be cut back significantly, this covers a lot of information about basic bone biology with no reference to OSM. Detailed information on OSM is introduced on Pg. 7 of the review. This manuscript would benefit from removing some detail and instead referring the reader to other published reviews. The text needs to refer to the scope of the review stated in lines 129-135.
Major points:
- Introduction: This section needs some adjusting to improve the clarity of the message conveyed, the information should be more focused to provide necessary details, not an overview of bone biology. Some of the flow between the sentences needs fixing, this might be achieved by re-ordering the introduction. (e.g. Why is it important first? (Line 65), it’s structure and then the cell types.) There are also quite a few instances of repetition. Some examples (not all) for improvement are below.
- Line 38: “Skeleton exists either as cortical bone” – The skeleton is the framework/structure, needs to be corrected to the “Skeleton contains cortical bone …and trabecular”
- Line 45: “mesenchymally derived cells” à “mesenchymal-derived cells”
- Line 45: “closely related to chondrocytes and adipocytes”. These too are mesenchymal derived. Need to specify.
- Line 44 and 47 – repetition
- Line 74, 86 – repetition
- Similar adjustments as mentioned in Point 1 need to also be applied to Section 2.
- The authors describe the differences between OSM signalling compared to LIF and also acting via LIFR. I think introduction and diagram of the two receptor complexes needs to come earlier, e.g Section 3 because this explains why there are comparisons frequently drawn with LIF signalling.
- Comparisons are drawn between human and mouse versions of OSM and LIF, yet the reader is unaware of how homologous these proteins are? Are the functional domains conserved? Could differences in those regions be causing the effects described?
- There are some instances where the authors discuss other IL6 family cytokines wrt signalling or osteoclasts formation, perhaps this could be delineated from the main text into a separate section towards the end.
- Line 342=345. This piece of information about OSM stimulating TRAP+ cells in human bone marrow cultures needs greater context/interpretation. Why is it important to mention this? How does this compare with the mouse studies?
- The review would benefit from a section which may discuss some of the limitations within the field, particularly these discrepancies between human and mouse, and calvaria vs bone marrow cells. Similarly a few comments about where the field should go next in terms of this research area would greatly enhance the review.
Minor points:
- Abstract, Line 18: “Enhanced trabecular bone” – in what way? Be specific – greater bone mass, change to bone quality?
- Acronym “BMC” Line 524 – typically within the bone field this acronym stands for “bone mineral content” – I would recommend switching it to “WBMC” – “whole bone marrow cell culture”
Author Response
1
Reviewer 1: De Souza et al provide a comprehensive review of how oncostatin M stimulates osteoclast formation. While there is thorough coverage of the literature related to OSM, substantial changes to the text are needed. Predominantly, Sections 1 and 2 need to be cut back significantly, this covers a lot of information about basic bone biology with no reference to OSM. Detailed information on OSM is introduced on Pg. 7 of the review. This manuscript would benefit from removing some detail and instead referring the reader to other published reviews. The text needs to refer to the scope of the review stated in lines 129-135.
Major points:
1. Introduction: This section needs some adjusting to improve the clarity of the message conveyed, the information should be more focused to provide necessary details, not an overview of bone biology. Some of the flow between the sentences needs fixing, this might be achieved by re-ordering the introduction. (e.g. Why is it important first? (Line 65), it’s structure and then the cell types.) There are also quite a few instances of repetition. Some examples (not all) for improvement are below.
The reason we have given quite an extensive introduction to the bone field (Introduction) and osteoclastogenesis (Osteoclast formation), is that our manuscript has been submitted to a special issue of IJMS with the title “OSM/OSMR and Interleukin 6 Family Cytokines in Physiological and Pathological Condition “. Since we assume that most people working in this field have only a limited knowledge in bone biology, we thought that we needed to give an overview of the bone and osteoclast field. Based upon the comments by reviewer, we have, however, now made extensive reductions in both the “Introduction” and “ Osteoclast formation” sections.
1. Line 38: “Skeleton exists either as cortical bone” – The skeleton is the framework/structure, needs to be corrected to the “Skeleton contains cortical bone …and trabecular”
We have made the suggested correction.
2. Line 45: “mesenchymally derived cells” à “mesenchymal-derived cells”
We have made the suggested correction.
2
3. Line 45: “closely related to chondrocytes and adipocytes”. These too are mesenchymal derived. Need to specify.
We are aware of the also chondrocytes and adipocytes are mesenchymal cells and thought the sentence indicated this. We have, however, now changed the text stating that osteoblasts, similar to chondrocytes and adipocytes are mesenchymal derived cells.
4. Line 44 and 47 – repetition
We thank the reviewer for alerting us to this repetition. The sentence on line 44 is now deleted to avoid this repletion.
5. Line 74, 86 – repetition
Many thanks for alerting us to this repetition. We have deleted the sentence on line 86.
2. Similar adjustments as mentioned in Point 1 need to also be applied to Section 2.
Please see our response to point 1 above, Introduction.
3. The authors describe the differences between OSM signalling compared to LIF and also acting via LIFR. I think introduction and diagram of the two receptor 2 complexes needs to come earlier, e.g Section 3 because this explains why there are comparisons frequently drawn with LIF signalling.
In response to this valuable comments we have now added a paragraph to the section “Expression of OSM, OSMR and LIFR in bone cells” introducing the similarities and differences between OSMR and LIFR. We have also added a new figure (Figure 3) to this part to demonstrate in which bone cells these receptors are expressed.
4. Comparisons are drawn between human and mouse versions of OSM and LIF, yet the reader is unaware of how homologous these proteins are? Are the functional domains conserved? Could differences in those regions be causing the effects described?
This information is included in the new paragraph added to the sections “Expression of OSM, OSMR and LIFR in bone cells”. We now describe the sequence alignment information between mOSM and hOSM and discussed how conserved domains in specific regions of the different ligands, both in mouse and human, interfere in the receptor specificity.
3
5. There are some instances where the authors discuss other IL6 family cytokines wrt signalling or osteoclasts formation, perhaps this could be delineated from the main text into a separate section towards the end.
We have taken out sentences mentioning IL-6 from previous part and now added a new section “Effects on bone resorption by other members of the IL-6 family” in which we summarize the knowledge on bone resorption induced by IL-6 and the other members besides OSM and LIF.
6. Line 342=345. This piece of information about OSM stimulating TRAP+ cells in human bone marrow cultures needs greater context/interpretation. Why is it important to mention this? How does this compare with the mouse studies?
Since the knowledge on stimulation of human osteoclasts by OSM is very limited we have deleted this piece of information.
7. The review would benefit from a section which may discuss some of the limitations within the field, particularly these discrepancies between human and mouse, and calvaria vs bone marrow cells. Similarly, a few comments about where the field should go next in terms of this research area would greatly enhance the review.
We have added some sentences in the “Concluding remarks” in response to these comments by the reviewer.
Minor points:
8. Abstract, Line 18: “Enhanced trabecular bone” – in what way? Be specific – greater bone mass, change to bone quality?
It is specified “enhanced trabecular bone mass” but to be more clear we have now added “caused by increased trabecular thickness”.
9. Acronym “BMC” Line 524 – typically within the bone field this acronym stands for “bone mineral content” – I would recommend switching it to “WBMC” – “whole bone marrow cell culture”
Since BMC is mentioned in rather few places we have now written “whole bone marrow cell culture” instead of using an acronym.
Reviewer 2 Report
The authors provided an overview on the role of Oncostatin M and Wnt16 in osteoclast formation.
There are some missing papers: Anjali Geethadevi, Cancer Res 2021; J Cell Biochem. 2020, Han Z; Leukemia. 2021,Newen G;
I would suggest the authors to focus one paragraph on the potential traslational impact of this axis in cancer especially for those cancers characterized by osteoclast activation.
I also suggest to summarize in a table the main findings focusing on the type of disease, role of oncostatin and the model used (vitro/vivo)
Author Response
Reviewer 2: The authors provided an overview on the role of Oncostatin M and Wnt16 in osteoclast formation.
There are some missing papers: Anjali Geethadevi, Cancer Res 2021; J Cell Biochem. 2020, Han Z; Leukemia. 2021, Newen G.
I would suggest the authors to focus one paragraph on the potential translational impact of this axis in cancer especially for those cancers characterized by osteoclast activation.
We thank the reviewer for the advice to include a discussion on the role of OSM in osteoclast formation induced by malignant tumors. Although there are several reports on OSM being pro- or anti-proliferative for different tumor cells, the interaction with bone cells in tumor metastasis is not much studied. When reading the suggested papers we found that:
- Anjali Geethadevi, Cancer Res 2021 is a paper on OSM and ovarian cancer describing that OSMR is expressed in ovarian cancer cells, that OSM is expressed in tumor-associated macrophages and that stimulation of OSMR in the tumor cells promotes proliferation. The paper does not include any studies on bone or bone cells.
- Han Z, J Cell Biochem 2020 is a paper in which the role miRNA for OSM in the RAW cell line 264.7 is studied and how overexpression of this miRNA suppress prostate cancer cell proliferation without any studies on bone or bone cells.
- Newen G, Leukemia 2021 is probably a paper by Bisht et al. in Leukemia 2021. This is a paper describing a role of OSM for the hematopoetic stem cell niche and the mobilization of hematopoetic stem cells in bone marrow without any studies on bone cells.
- In the latest issue of J Clin Invest, Araujo et al. also have published an interesting study showing that OSM expressed by myeloid cells present in breast cancers induces an enhanced tumorogenic phenotype of tumor stromal fibroblast, also arguing for an important role of OSM in breast cancer, but no studies on effects on bone cells in the vicinity of tumor metastasis were performed.
We have found two papers with a relation to OSM, cancer and bone cells:
- Bolin et al. Genes and Cancer 2012 describing the expression of OSM in estrogen-positive and estrogen-negative breast cancer cells and the role of OSM for the loss bone caused by these cells.
- Omokehinde et al. J Bone Min Res 2021 which show that overexpression of OSM in breast cancer cells increase their dissemination. Although no studies on bone were included, this paper supports the finding by Bolin et al.
We have difficulties to understand why the three studies listed by the reviewer should be included in a paper describing effects by OSM on osteoclast formation and bone resorption. We have,however, included the observations by Bolin et al. and Omokehinde et al. in the section “Concluding remarks”.
I also suggest summarizing in a table the main findings focusing on the type of disease, role of oncostatin and the model used (vitro/vivo).
Even though OSM has been suggested to regulate bone mass in some inflammatory skeletal diseases, as mentioned in the manuscript, we have not been able to find other studies investigating a pathogenetic role by OSM on osteoclasts and bone resorption in different diseases. We are therefore unable to make such a table.
Round 2
Reviewer 1 Report
The authors have addressed the comments previously raised and improved the manuscript.
Minor points that should be corrected during proofing:
- Pg. 2 - removal of the phrase "terminally differentiated", osteoblasts can differentiate into osteocytes and lining cells.
- Pg. 8 - "immune histochemical", more commonly referred to as "immunohistochemistry". Switching it to the latter, will be clearer for readers.
- Pg. 11. - "Lif had not effect on any of the isoforms." Needs to be "affect".